

# Multi skin lesions classification using fine-tuning and data-augmentation applying NASNet

Elia Cano, José Mendoza-Avilés, Mariana Areiza, Noemi Guerra, José Longino Mendoza-Valdés and Carlos A. Rovetto

Computer Science, Universidad Tecnológica de Panamá, Panama, Panama

## ABSTRACT

Skin lesions are one of the typical symptoms of many diseases in humans and indicative of many types of cancer worldwide. Increased risks caused by the effects of climate change and a high cost of treatment, highlight the importance of skin cancer prevention efforts like this. The methods used to detect these diseases vary from a visual inspection performed by dermatologists to computational methods, and the latter has widely used automatic image classification applying Convolutional Neural Networks (CNNs) in medical image analysis in the last few years. This article presents an approach that uses CNNs with a NASNet architecture to recognize in a more accurate way, without segmentation, eight skin diseases. The model was trained end-to-end on Keras with augmented skin diseases images from the International Skin Imaging Collaboration (ISIC). The CNN architectures were initialized with weight from ImageNet, fine-tuned in order to discriminate well among the different types of skin lesions, and then 10-fold cross-validation was applied. Finally, some evaluation metrics are calculated as accuracy, sensitivity, and specificity and compare with other CNN trained architectures. This comparison shows that the proposed system offers higher accuracy results, with a significant reduction on the training paraments. To the best of our knowledge and based in the state-of-art recompiling in this work, the application of the NASNet architecture training with skin image lesion from ISIC archive for multi-class classification and evaluated by cross-validation, represents a novel skin disease classification system.

# INTRODUCTION

Skin cancer is frequent in the USA, Australia, and Europe (*Codella et al., 2016*) with 20% of Americans developing this kind of disease by the age of 70, 4% of all cancers in Asians, 5% in Hispanics and an annual cost of treating in the U.S. estimated at $8.1 billion, signifying skin cancer as a severe public health problems (*Skin Cancer Foundation, 1981*).

Mutations in the DNA of epidermal cells lead to out-of-control, abnormal growth and cause skin cells to multiply rapidly forming a malignant tumor, the main causes of that mutation are harmful ultraviolet (UV) rays and the use of UV tanning machines. The chief types of skin cancer are squamous cell carcinoma (SCC), basal cell carcinoma (BCC), Merkel cell carcinoma (MCC) and melanoma (MEL). Despite how deadly skin cancer is,

Corresponding author
Carlos A. Rovetto,
carlos.rovetto@utp.ac.pa

with a 5-year survival rate, it can be up to 99% if diagnosed and treated early enough; however, a delayed diagnosis causes a decrease to 23% in the survival rate.

The diagnosis of skin cancer starts with a visual inspection of a dermatologist. Due to the nature of some lesion's types, a correct diagnosis is important (*Haenssle et al., 2018*), although the accuracy of the diagnosis is correlated to the professional experience. Using dermatoscopic images and visual inspection, dermatologists can achieve an accuracy of 75–84% (*Fabbrocini et al., 2011*; *Ali & Deserno, 2012*). Finally, biopsies can detect the malignancy of a skin growth, but they are also the most invasive techniques. This process takes approximately 15 min. Despite being a short time, technological advances open the possibility of improving accuracy while reducing time and costs through image analysis. This work gives an approach to increase the accuracy and reduce the time response in non-invasive way.

Automatic image classification using Convolutional Neural Networks (CNN) has been widely used in the analysis of medical images (*Litjens et al., 1995*; *Lo et al., 1995*); however, until 2012 with the work of *Krizhevsky, Sutskever & Hinton (2012)* using the AlexNet architecture that increases the work used by these models for the classification of images through CNNs. Other works have used convolutional architectures with important results (*Fukushima, 1980*; *Deng et al., 2009*; *Russakovsky et al., 2015*; *Hembury et al., 2003*; *Simonyan & Zisserman, 2015*; *Szegedy et al., 2014*; *Xie et al., 2017*).

Deep learning technique was applied in the search for automated lesion classification for unifying the dermatologist's professional experiences and supporting them in the diagnosis, convolutional neural network (CNN) training for the detection and classification of skin diseases is carried out using a set of data. This uses highly standardized dermatoscopic images that are acquired through a specialized instrument or histological images acquired through invasive biopsy and microscopy. To train the network, some authors operate with datasets from open-access dermatology repositories, others with repositories belonging to hospitals or clinics where samples are taken (*Haenssle et al., 2018*; *Rivenson et al., 2018*), or a combination of the previous two. On the other hand, the images used for diagnosis can be taken using digital cameras (*Połap et al., 2018*; *Nasr-Esfahani et al., 2016*; *Amelard et al., 2015*; *Alcón et al., 2009*; *Cavalcanti, Scharcanski & Baranoski, 2013*; *Cavalcanti & Scharcanski, 2011*; *Chao, Meenan & Ferris, 2017*; *Zoph et al., 2018*) or the camera of a smartphone.

The authors in *Haenssle et al. (2018)* trained a Inception v4 architecture to perform a binary classification between of melanoma and benign skin lesions. The results were compared with the opinion of 58 dermatologists with different levels of experience. The article outcome is especially valuable because it offers proof of the importance of computer aid, independent of the physician experience level.

One of the most significant advances in the field of skin disease classification and detection comes with the creation of ISIC Challenge in 2016 (*Codella et al., 2016*). ISIC publishes the largest skin disease data set, divided by 14 classes. Some of the classes were merged or omitted as a result of the small number of images on it.

Since 2016 CNN was used in skin lesions classifications with several approaches based on the number of classes to classify (binary, multi-class), the way the CNN is used

(Feature Extractor, end-to-end training and learning from scratch) (*Brinker et al., 2018*) and some use the segmentation with a U-net before feeding the training model.

A CNN is used as a feature extractor when is pre-trained with a large dataset (ImageNet) and the fully connected layer is removed—usually, the data is augmented and normalized. In most of the papers found with this approach, AlexNet architecture is used as a feature extractor with a K-neighbor or support vector machine as a classifier.

The images used to classify could come from their own source as in *Pomponiu, Nejati & Cheung (2016)* with 399 photos taken with a standard camera—achieving a sensitivity of 92.1%, specificity of 95.18% and an accuracy of 93.64%. In contrast, in *Kawahara, Bentaieb & Hamarneh (2016)* and *Codella et al. (2015)* use a public libraries as DermoFit (*Edinburgh Innovations, 2019*) and ISIC dataset respectively.

The authors in *Kawahara, Bentaieb & Hamarneh (2016)* present a multi-classification performed by logistic regression with a final accuracy of 81.8%—the data is splinted in validation, training and test. On the other hand, two-fold cross-validation is used in *Codella et al. (2015)* for two task of binary classification—melanomas vs non-melanoma (accuracy of 93.1%) and melanoma vs atypical nevi (73.9% accuracy).

A widely used approach is transfer learning, were an architecture is initialized with the weight of another data and fine-tuned to fit the new dataset. As in *Esteva et al. (2017)* where the authors train a Inception v3 with 129,450 images from a private source and 3,374 images obtained from dermatoscopic devices. Two problems of binary classification were tested, keratinocyte carcinomas vs benign seborrheic keratosis and malignant melanomas vs benign nevi. The outcome was present with the Area Under the Receiver Operating Characteristics metric—0.96 for carcinomas and 0.94 for melanomas.

The authors in *Han et al. (2018)* proposed to train a ResNets for a multi-classification of 12 skin lesions tested with the ASAN dataset (Average AUC 0.91 ± 0.01, sensitivity 86.4 ± 3.5 and specificity 85.5 ± 3.2) and DermoFit (*Edinburgh Innovations, 2019*) (Average AUC 0.89 ± 0.01, sensitivity 85.1 ± 2.2 and specificity 81.3 ± 2.9). Residual neural network is also used in (*Bi et al., 2017*) to evaluate three approaches multi-class classification (Melanoma, Seborrheic keratosis and Nevus), binary classification (Melanoma vs Seborrheic keratosis and Nevus), and the ensemble approach. The latter approach got the best AUC results with Melanoma 85.40, Seborrheic keratosis 97.60 and average of 91.50 with the ISIC 2017 Challenge dataset.

Other architecture implemented with transfer learning is the VGGNet. One of the first examples of this is shown in *Sun et al. (2016)* with DermQuest archive classifying among 198 classes (accuracy 50.27%). After, the authors *Romero Lopez et al. (2017)* modified VGGNet train with the ISIC 2016 Challenge dataset to discriminate between malignant and benign. The best configuration achieved an accuracy of 81.33%, sensitivity 0.78 and precision 0.79.

In this paper, NASNet architecture is implemented to recognize 8 skin diseases more than of the majority of previous cases. We can identify three types of cancer: Squamous Cell Carcinoma, Melanoma and Basal Cell Carcinoma. Also, our model can discriminate nevus, the most common kind of mole, of the Melanomas.

The rest of the article is organized as follows: materials and methods are explained in "Materials and Methods". In "Results" the results and discussion are presented. Finally, the conclusions and future works are given in "Discussion".

## MATERIALS AND METHODS

A formal statement of the problem, from an example-based learning problem or supervised learning problem, is the following one:

In this work a Softmax function is used, therefore, the equation

$$a^l = \sigma(z^l) \tag{1}$$

could be rewritten as

$$\sigma(z^l) = \frac{e^{z^l}}{\sum_{j=1}^{K} e^{z^j}} \tag{2}$$

being $\Sigma^{z^l}$ a probability distribution that will center around the positions of the values, applying it to the largest entries (*Nwankpa et al., 2018*).

Let $X$ and $Y$ be two metric spaces: $X$ (skin image), $Y$ (corresponding class label) and a (target) function $y: X \rightarrow Y$, specified only in the finite aggregate of points: $y(X^1),\ldots, y(X^8)$, that is, the labels of objects $X^1,\ldots, X^8$ are known (*Muhamedyev, 2015*). Where $X$ is split to classes according to the skin disease ["AK", "BCC", "BKL", "DF", "MEL", "NV", "SCC", "VASC"]; after one-hot encoding was applied the skin disease classes could be noted as $H_1,\ldots, H_8$, where

$$H_i\{x \in X | y(x) = i\} \text{ at } i \in \{1,\ldots,8\}: \quad X = \bigcup_{i=1}^{8} H_i \tag{3}$$

The target function $y: X \rightarrow Y$, that discriminates well among the different class labels, is describes as the working process of a neural network in

$$a_j^l = \sigma\left(\sum_{k=1} w_{jk}^l a_k^{l-1} + b_j^l\right) \tag{4}$$

with these notations; the vector activation's components $a_j^l$ are represented as the sum over all neurons $k$ in the $(l-1)$ layer in a matrix form, where the weight from each layer $l$ defines as $w^l$ with $j$ and $k$ are the representation of row and columns, respectively. The components of the bias vector are just the values $b_j^l$. Equation also can be rewritten in a compact vectorized

$$a^l = \sigma\left(\sum_m w^l a^{l-1} + b^l\right) \tag{5}$$

also written in terms of the weighted input, as *Nielsen (2015)*

$$a^l = \sigma(z^l) \tag{6}$$

In order to quantifies the error between predicted values $\hat{y}$ and expected values $y$ a cost function $J(\hat{y}, y)$ is applied.

$$J(\hat{y}, y) = -\sum_{j=0}^{M} \sum_{i=0}^{N} \left( y_{ij} * \log(\hat{y}_{ij}) \right) \tag{7}$$

where the output of the function is given by

$$J(\hat{y}, y) = \{1, \ \hat{y} \neq y \ \text{ or } \ 0, \hat{y} = y\} \tag{8}$$

The way to decrease an objective function $J(\theta)$ parameterized by the parameters of a

$$\{\theta = (\hat{y}, y) | \theta \in \mathbb{R}^d\} \tag{9}$$

through the gradient descent model by updating the parameters in the opposite direction of the gradient of the objective function $\nabla_\theta J(\theta)$ with respect to the parameters. The learning rate $\eta$ determines the size of the steps we take to reach a (local) minimum. Depending on the amount of data, we trade off the accuracy of the parameter update and the time it takes to perform an update. Thus, updated is perform stochastic gradient descent (SGD) for every example $x^{(i)}$ and label $y^{(i)}$ for every mini batch of $n$ training examples (*Ruder, 2016*)

$$\theta = \theta - \eta \cdot \nabla_\theta J\left(\theta; \ x^{(i:i+n)}; \ y^{(i:i+n)}\right) \text{ for } n = 30 \tag{10}$$

In other words, the parameters updated was made by feeding the model with mini batches of 45 images—this amount of batches prevent an overload on the GPU memory. It take for the computer 975 steps to update all the parameters, this process was repeated until the error stop to decrease. The mini-batches method is chosen because reduce the variance of parameter updates, which can lead to more stable convergence; it can make use of highly optimized matrix optimizations common to the state-of-the-art deep learning libraries that make computing the gradient with respect to a very efficient mini-batch; and reduce the stored examples on the computer's RAM.

Adaptive Moment Estimation (ADAM) was applied to calculate the learning rate $\eta$ and store an exponentially decreasing average of the past gradients

$$m_t = \beta_1 m_{t-1} + (1 - \beta_1) g_t \tag{11}$$

and the past square gradients

$$v_t = \beta_2 v_{t-1} + (1 - \beta_2) g_t^2 \tag{12}$$

*Ruder (2016)* those are estimates of the first momentum (the mean) and the second moment (the uncentered variance), respectively. The $m_t$ is where the past normalized gradient is recorded, called the first moment, and $v_t$ refers to the adaptive gradient decrease

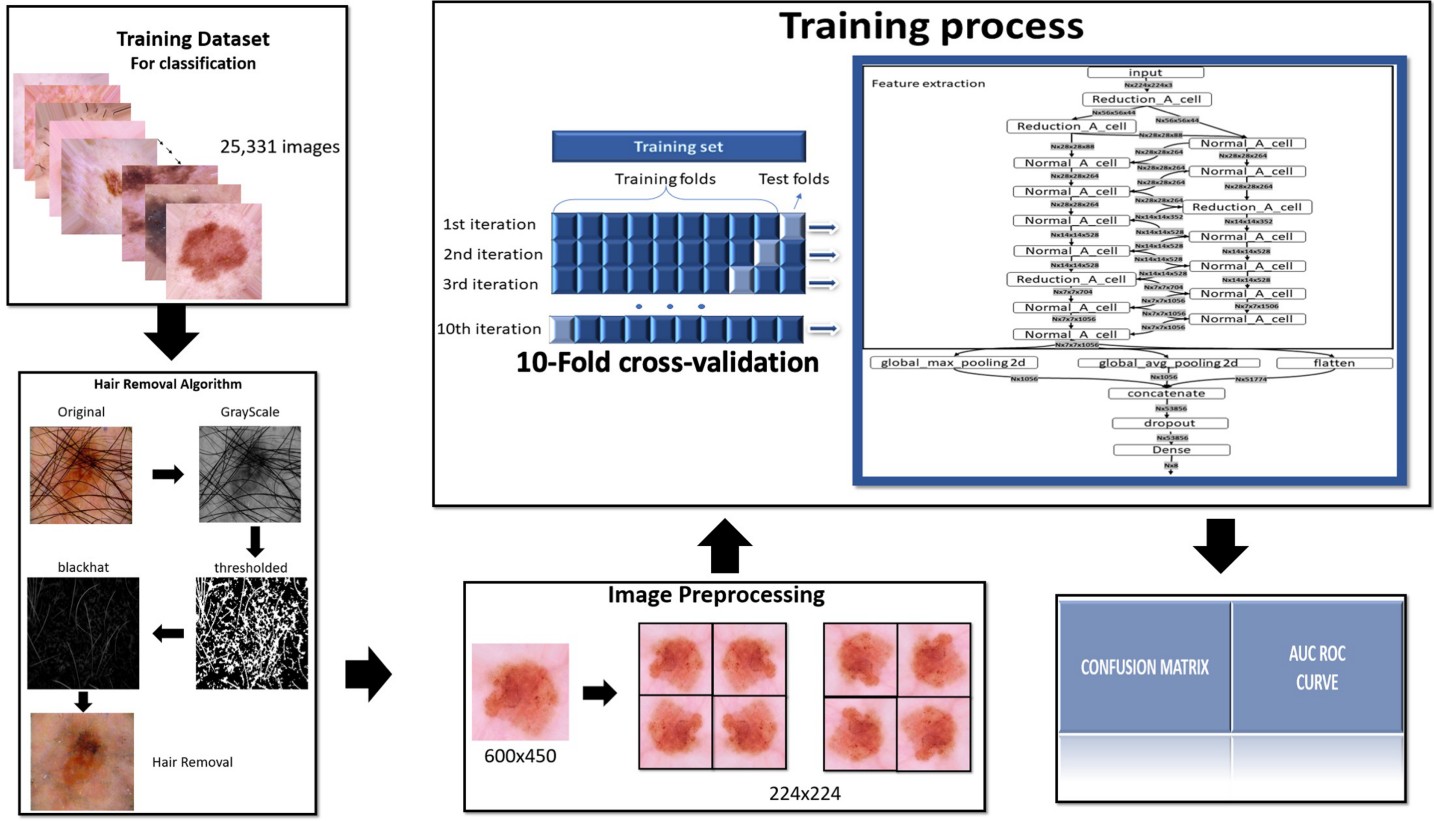

**Figure 1 Methodology.** To complete this project is used the methodology summarized in this figure.

displayed in the RMSprop (*Ruder, 2016*), which in turn is called the second moment. As the authors of Adam explain, $\beta_1$ and $\beta_2$ refer to the decomposition rate, which are small due to the initial time steps, this causes them to be biased towards zero (*Kingma & Ba, 2015*). Where $g_t$ denotes the gradient at time step $t$. $g_{t,i}$ is then the partial derivative of the objective function w.r.t. to the parameter $\theta_i$ at time step

$$t : g_{t,i} = \theta_{\nabla} J(\theta_{t,i}) \tag{13}$$

ADAM performance akin to other optimizers as RMSprop, Adadelta in similar circumstances. In *Kingma & Ba (2015)* shows Adam to slightly outperform RMSprop due to bias correction when optimization is ending and as gradients become more scattered. In the measure, Adam might be the best overall choice.

## Workflow of the proposed systems

To complete this project is used the methodology summarized in the following Fig. 1.

## Dataset

The dataset used for this project comes from the ISIC Training Challenge 2019 (*Tschandl, Rosendahl & Kittler, 2018*; *Codella et al., 2018*; *Combalia et al., 2019*). This dataset consists

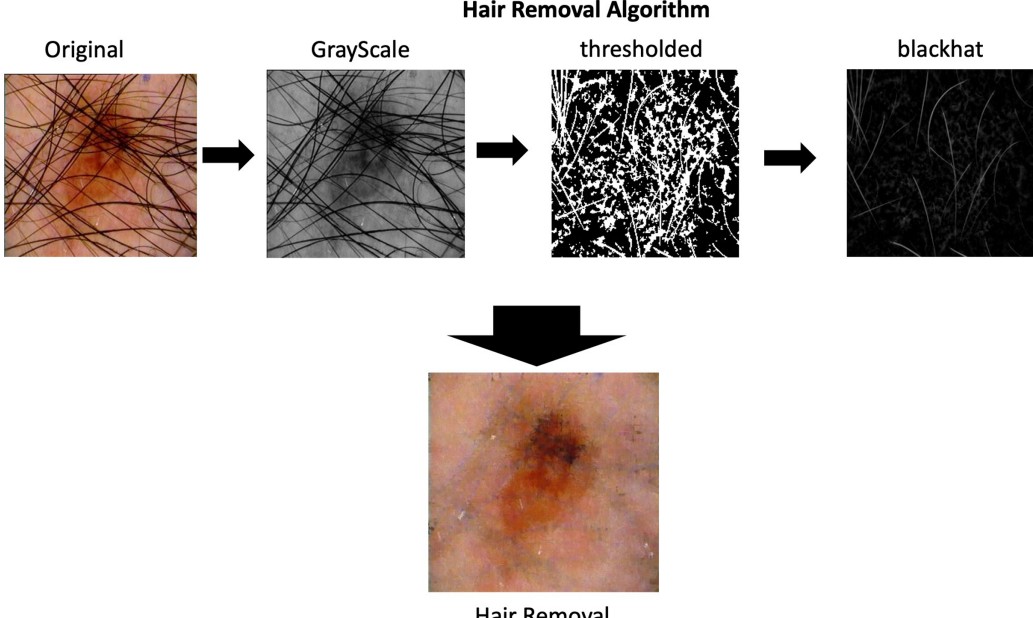

**Figure 2 Hair removal process.** Hair removal process applied in one of the images from ISIC dataset.

in 25,331 JPEG images of skin lesions, divided in Actinic Keratosis (AK), Squamous cell carcinoma (BCC), Benign keratosis (solar lentigo/seborrheic keratosis/lichen planus-like keratosis) (BKL), Dermatofibroma (DF), Melanoma (MEL), Melanocytic nevus (NV), Squamous cell carcinoma (SCC) and Vascular lesion (VASC).

## Hair removal

As part of the process of image pre-processing is imperative to remove the hair that appears in skin lesions images. The algorithms for hair removal on skin images have been widely studied, the most simple and efficient one is then carried out by *Lee et al. (1997)* called DullRazor. This algorithm identifies the dark hair location through generalized grayscale morphological closing operation, after the hair pixels shape is checked, they are replaced using bilinear interpolation and smooth by an adaptive median filter.

The complete process is depicting in the Fig. 2 on the hair removal section.

## Data augmentation

Due to the class imbalance Data Augmentation is applied. This technique is used to increase the amount of data available in the classes. The proper use of this technique increases the generalization of the model. It also prevents overfitting, since increasing the number of variations in the data brings it closer to reality. With this additional data the model can learn, during training, properties such as contrast invariance, location invariance, rotational invariance, and the like.

Data Augmentation settings applied to the dataset are described in Table 1 and the effects would be seen in the Fig. 3. These methods are applied through the keras

Table 1 **Augmentation details.** Data Augmentation settings applied to the dataset are described in this table and the effects would be seen in the Fig. 3. These methods are applied through the keras library with the ImageDataGenerator function. The process is also used to resize images to 224 × 224.

| Augmentation | Percentage or ratio range |
|---|---|
| rotation_range | 180 |
| width_shift_range | 0.1 |
| height_shift_range | 0.1 |
| zoom_range | 0.1 |
| horizontal_flip | true |
| vertical_flip | true |
| fill_mode | nearest |

# Data-Augmentation

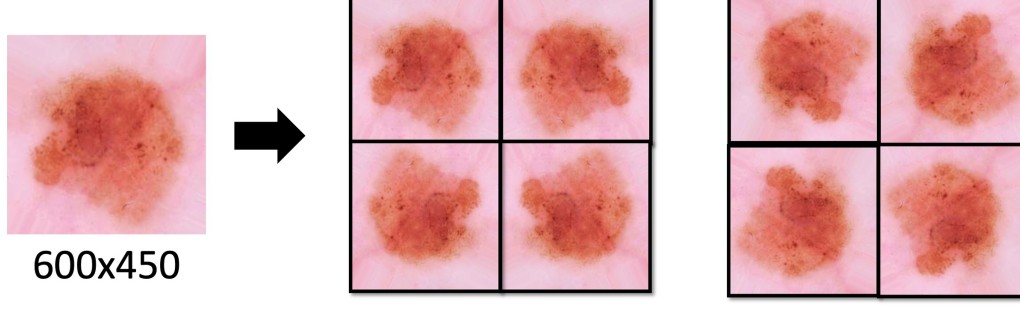

Figure 3 **Data-Augmentation.** Data-Augmentation applied in one of the images from ISIC dataset.

library with the ImageDataGenerator function. The process is also used to resize images to 224 × 224.

Some of the parameters that we pass through are:

rotation_range: range of degrees for random rotations

width_shift_range: the fraction of the total width that the image can be shifted by

height_shift_range: the fraction of the total height that the image can be shifted by

zoom_range=0.1: represents the fraction of the image that can be zoomed in or out

horizontal_flip=True: randomly flips the input horizontally

vertical_flip=True: randomly flips the input vertically

fill_mode="nearest": the specification to fill points outside the input limits.

## Neural network architectures

The NASNet architecture is a convolutional neural network developed by an IA created by the Google Team in 2018. Their authors said, "Our model is 1.2% better in top-1 accuracy than the best human-invented architectures while having 9 billion fewer FLOPS—a

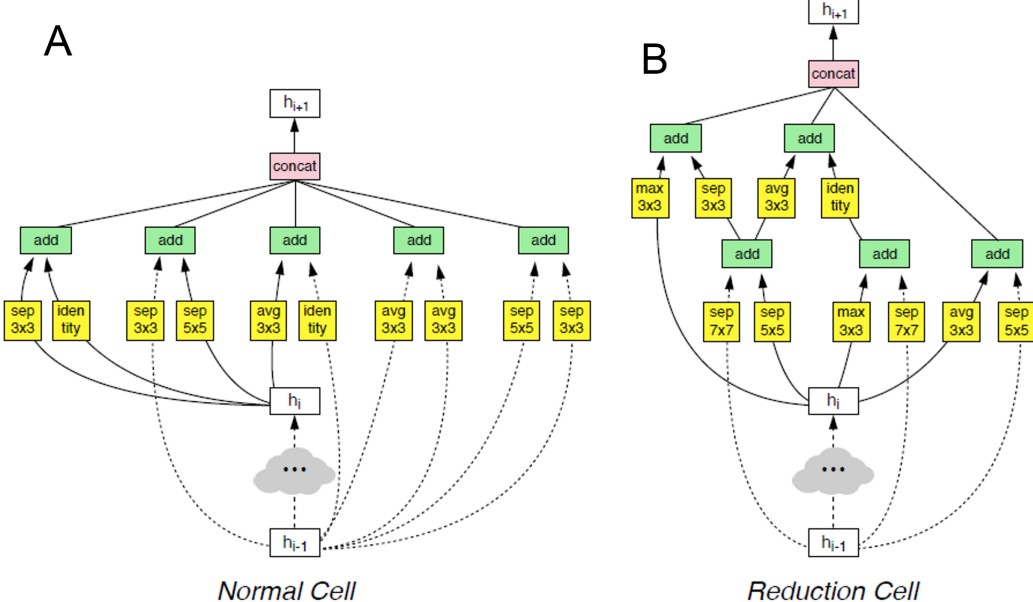

**Figure 4 The two main functions of NASNet architecture.** This architecture is composed of convolutional cells. The two main functions are (A) Normal Cell and (B) Reduction Cell, shown in this figure. The Reduction Cell returns a feature map height and width reduced by a factor of two. On the other hand, the Normal Cell returns a feature map with normal cells with the same input dimensions. The model used for this purpose was NASNet-A (4 @ 1056), where the number 4 represents the number of cell repeats, and 1,056 corresponds to the number of filters in the penultimate layer of the network.

reduction of 28% in computational demand from the previous state-of-the-art model". Due to the accuracy increase registered in the state-of-the-art and reduction of the computational demand, this architecture is applied in this project as a feature extractor (*Zoph et al., 2018*).

This architecture is composed of convolutional cells. The two main functions are Normal Cell and Reduction Cell, shown in Fig. 4. The Reduction Cell returns a feature map height and width reduced by a factor of two. On the other hand, the Normal Cell returns a feature map with normal cells with the same input dimensions. The model used for this purpose was NASNet-A (4 @ 1056), where the number 4 represents the number of cell repeats, and 1056 corresponds to the number of filters in the penultimate layer of the network.

At the end of the last Normal_A_cell, global_max_pooling2d, global_avg_pooling2d, and flatten layers were fed. The dimensions of the tensors were reduced to Nx1056 in the first two layers, where the number of images the system is trained is noted as N. When we applied the filters, we found the largest number in the global_max_pooling2d layer and obtained the arithmetic average in global_avg_pooling2d. The output of the flatten layers was Nx51774. The outputs of these three layers were concatenated in the concatenate layer. This layer fed the dropout layer, which turned off some neurons to prevent overfitting of the network. Finally, it fed the dense layer, which offered the inference of the model through a $\sigma(z^l)$ function. These steps are depicted in Fig. 5.

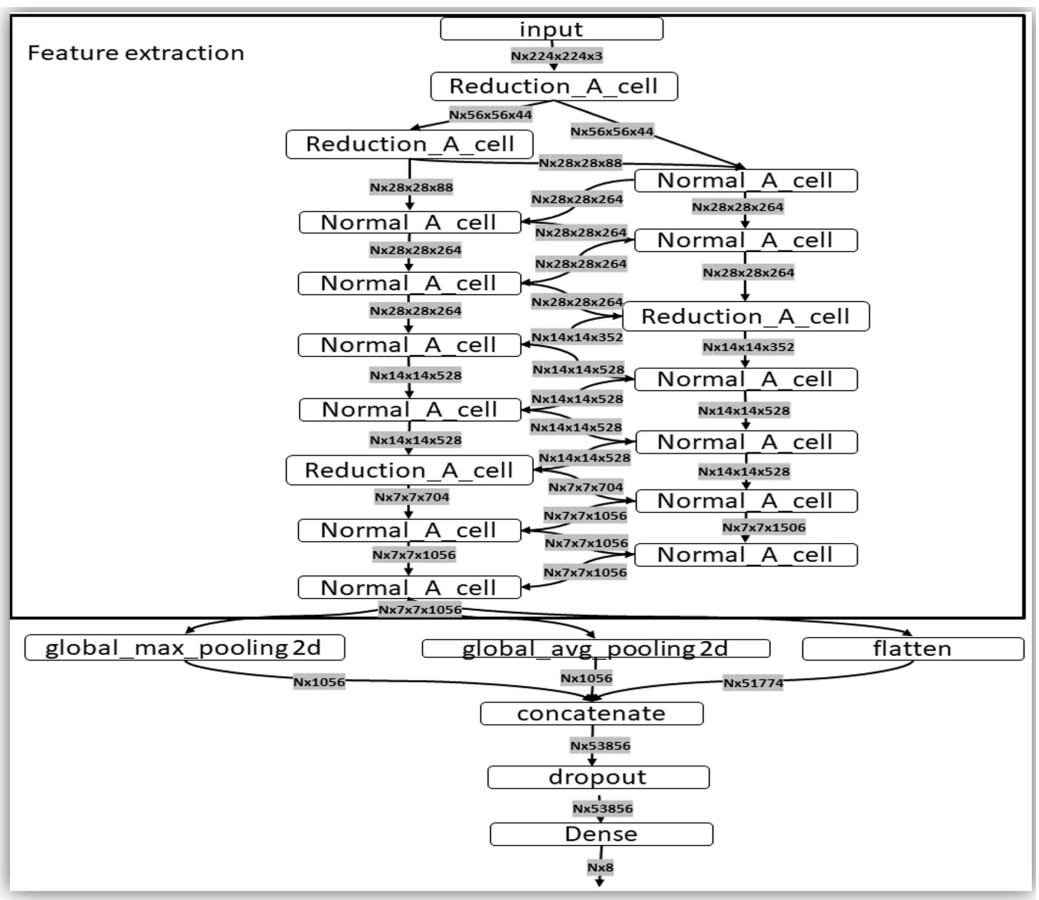

**Figure 5 Representation of the steps of the extraction feature map and classification.** This layer fed the dropout layer, which turned off some neurons to prevent overfitting of the network. Finally, it fed the dense layer, which offered the inference of the model through a $\sigma(z^l)$ function. These steps are depicted in this figure.

## Evaluation metrics

Through the confusion matrix we can obtain the productivity of the model during the training or the development of a classification problem. This shows us in detail how many times the model is wrong when making predictions. The number of correct and incorrect predictions is obtained by counting values and separated them from each class. It gives us an idea not only of the mistakes that the model makes but of the types of mistakes it makes. This matrix allows us to measure Recall, Precision, Accuracy, and the AUC-ROC curve. This matrix describes the complete performance of the model. Table 2 shows the data distribution for multiclass classification.

This matrix is composed of:

True Positive (TP): the observation is positive and was predicted to be positive.
True Negative (TN): the observation is negative and was predicted to be negative.
False Positive (FP): the observation is negative, but it was forecast as positive.
False Negative (FN): the observation is positive, but it was predicted as negative.

**Table 2 Data distribution for the multiclass classification confusion matrix.** Through the confusion matrix we can obtain the productivity of the model during the training or the development of a classification problem. This shows us in detail how many times the model is wrong when making predictions. The number of correct and incorrect predictions is obtained by counting values and separated them from each class. It gives us an idea not only of the mistakes that the model makes but of the types of mistakes it makes. This matrix allows us to measure Recall, Precision, Accuracy, and the AUC-ROC curve. This matrix describes the complete performance of the model. This table shows the data distribution for multiclass classification.

| Known Class | 0 | 1 | 2 | … | J |
|---|---|---|---|---|---|
| 0 | **TP** | FN | FN | FN | FN |
| 1 | FP | TN | FN | FN | FN |
| 2 | FP | FN | TN | FN | FN |
| … | FP | FN | FN | TN | FN |
| J | FP | FN | FN | FN | TN |

From the confusion matrix we compute the accuracy. It is the ratio of the number of correct predictions to the total number of input simples. Accuracy $= \frac{TP+TN}{TP+TN+FP+FN}$. It works well only if there are equal numbers of samples belonging to each class.

Recall gives us the number of correct positive results divided by the number of all relevant samples, where all samples should have been identified as positive. Recall $= \frac{TP}{TP+FN}$.

Precision is a measure of exactness. It defines the probabilities of the number of correct positive results to the number of positive results predicted by the classifier. Precision $= \frac{TP}{TP+FP}$.

We use an F1 Score to know how precise and robust our classifier is. F1 Score is a balance between precision and recall. The range for the F1 Score is [0, 1]. The greater the F1 Score the better the performance of our model.

$$\text{F1 Score} = 2 * \frac{\text{Recall} * \text{Precision}}{\text{Recall} + \text{Precision}}$$

To show the performance of the classification model in all the classification thresholds, we use the curve's ROC. This curve is plotted with the True Positive Rate TPR $= \frac{TP}{TP+FN}$ against the False Positive Rate FPR $= \frac{FP}{FP+TN}$, where TPR is on the $y$-axis and FPR on the $x$-axis.

The area under one of the ROC curves can be used as a measure of accuracy in many applications and is called the precision of surface-based measurement. Also, the ROC graph contains all the information contained in the matrix of errors (*Nielsen, 2015*).

## Computer characteristics used in the classification of images

We use a computer with high computational power for the analysis of large quantities of images. CUDA cores are used for to obtain better performance with the TensorFlow and Python library. These cores are owned by NVIDIA brand video cards. Due to this data, an inspection of the models and specifications of the most current cards was carried out.

The selected card is the NVIDIA RTX 2080TI with 11 GB of video memory which contains 4352 CUDA cores, incorporates the Turing architecture and brings the Deep

**Table 3 Computer characteristics used in the classification of images.** An Intel Core i9-7900X processor with 10 cores at 3.3 GHz, 64 GB of RAM at 3,600 MHz, a 2 TB SSD and a 4 TB HDD. These characteristics are describing below.

| Component | Description |
| --- | --- |
| Power Supply | Cooler Master Watt Maker 1500–1500 W |
| Mother board | Asus ROG STRIX X299-E GAMING—LGA2066 |
| Chip | Intel Core i9-7900X a 3.3 GHz (Skylake-X) |
| RAM | 64 GB 3600 MHz |
| GPU | GeForce RTX 2080 Ti GDDR6 (×2) NVLINK 11 GB RAM |
| SSD | 2 TB |
| HDD | 4 TB |

Learning Super Sampling (DLSS) technology which includes the Core Tensor. This is an artificial intelligence engine of 114 TFLOP power, which makes the card the one indicated for work on the project. In addition, it has the connection between NVLINK graphics cards that increases the capacity and speed of analyzing data between 5 to 10 times faster than other graphics cards and has a transfer power of 100 GB/s. An Intel Core i9-7900X processor with 10 cores at 3.3 GHz, 64 GB of RAM at 3600 MHz, a 2 TB SSD and a 4 TB HDD. These characteristics are describing below in Table 3.

## RESULTS

In summary, the proposed algorithm was training, validated, and tested in 25,331 images of skin lesions divided into eight classes, taken from the ISIC 2019 File. All images were resized to a size of 224 × 244 using bi-linear interpolation, normalized and data augmented to manage the unbalance between classes as in *Shi & Malik (2000)*.

Multi-class skin lesion classification comes with the problem of severe class imbalance and the small size of those themselves currently available, which represents a challenge for training purposes, therefore, data-augmentation is applied to avoid any bias and overfitting.

The best results with the proposed method were achieved with the following tuning. Firstly, a weighted Cross-entropy is used as loss function to estimate the parameters of all deep models. In addition, the Adam optimizer is initializing with a learning rate of 0.0001 and then it is reduced by 20% if the validation lose function does not decrease by 0.0001 every 45 iterations. Finally, early stopping stops the learning process when the F1-score is not increased by 0.001 through each 45 iterations used to avoid the overfitting that may occur before the convergence of deep models as well as speed up the learning procedures. Thus, the overfitting is prevented, and the bias is reduced. The implementation is carrying out with TensorFlow and Keras libraries. The tuning previous mentioned is also applied to train other CNN architectures.

The dataset is splitting based on 10-fold cross-validation. Usually deep learning workstation use the library sklearn to split in folds the original dataset and run until the end of the model training without any human interference. Nevertheless, due to

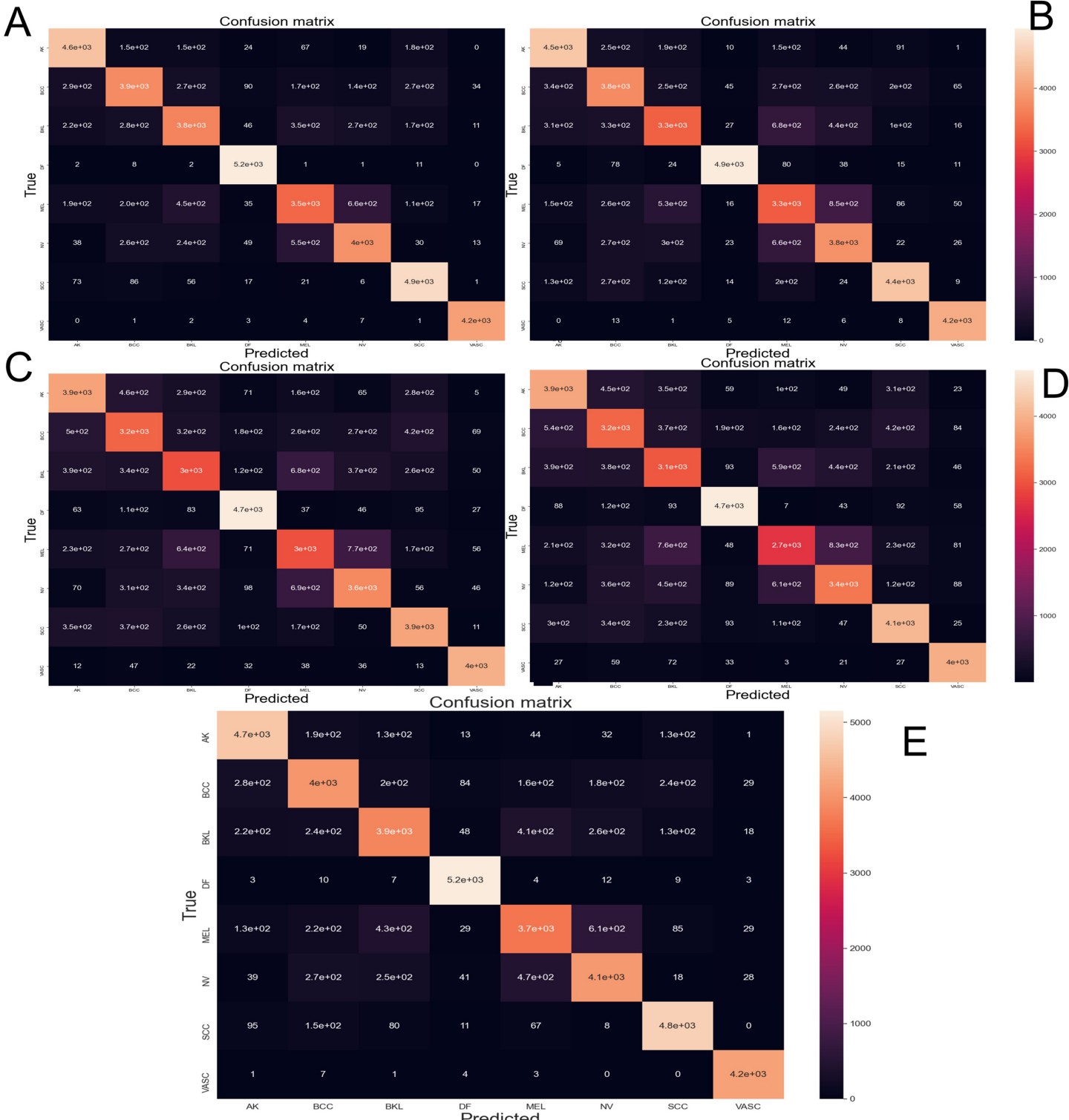

**Figure 6 Confusion matrix yielded by the proposed CNN method.** (A) NasnetMobile, (B) Inception V3, (C) InceptionResNetV2, (D) Dense-Net201 and (E) Xception. Where Actinic keratosis (AK), Basal cell carcinoma (BCC), Dermatofibroma (DF), Melanoma (ML), Nevus (NV), Pigmented benign keratosis (PBK), Seborrheic keratosis (SK), Squamous cell carcinoma (SL), Vascular Lesion (VL).

**Table 4 Computer characteristics used in the classification of images.** Through the years several studies have been conducted in this field. A continuation (below table) comparison between previous studies and our proposed method is presented.

| Diseases | NasnetMobile | | | Inceptionv3 | | | InceptionResNetV2 | | | DenseNet201 | | | Xception | | |
|---|---|---|---|---|---|---|---|---|---|---|---|---|---|---|---|
| | Precision | Recall | F1-score | Precision | Recall | F1-score | Precision | Recall | F1-score | Precision | Recall | F1-score | Precision | Recall | F1-score |
| AK | 0.87 | 0.82 | 0.85 | 0.86 | 0.82 | 0.84 | 0.74 | 0.71 | 0.72 | 0.74 | 0.70 | 0.72 | 0.90 | 0.86 | 0.88 |
| BCC | 0.71 | 0.80 | 0.75 | 0.72 | 0.72 | 0.72 | 0.61 | 0.63 | 0.62 | 0.61 | 0.61 | 0.61 | 0.78 | 0.79 | 0.78 |
| BKL | 0.72 | 0.75 | 0.74 | 0.63 | 0.70 | 0.67 | 0.57 | 0.60 | 0.59 | 0.59 | 0.57 | 0.58 | 0.74 | 0.78 | 0.76 |
| DF | 0.99 | 0.94 | 0.97 | 0.95 | 0.97 | 0.96 | 0.91 | 0.88 | 0.89 | 0.90 | 0.89 | 0.89 | 0.99 | 0.96 | 0.97 |
| MEL | 0.68 | 0.73 | 0.70 | 0.63 | 0.61 | 0.62 | 0.58 | 0.60 | 0.59 | 0.52 | 0.63 | 0.57 | 0.70 | 0.76 | 0.73 |
| NV | 0.76 | 0.78 | 0.77 | 0.74 | 0.70 | 0.72 | 0.69 | 0.69 | 0.69 | 0.65 | 0.67 | 0.66 | 0.78 | 0.79 | 0.79 |
| SCC | 0.94 | 0.84 | 0.89 | 0.85 | 0.89 | 0.87 | 0.75 | 0.75 | 0.75 | 0.78 | 0.74 | 0.76 | 0.92 | 0.89 | 0.90 |
| VASC | 0.99 | 0.98 | 0.99 | 0.99 | 0.96 | 0.97 | 0.95 | 0.94 | 0.95 | 0.94 | 0.91 | 0.92 | 1.00 | 0.97 | 0.99 |
| Accuracy | | | 0.83 | | | 0.79 | | | 0.72 | | | 0.71 | | | 0.85 |
| Macro Avg | 0.83 | 0.83 | 0.83 | 0.80 | 0.80 | 0.80 | 0.73 | 0.72 | 0.72 | 0.72 | 0.71 | 0.71 | 0.85 | 0.85 | 0.85 |
| Weighted Avg | 0.84 | 0.83 | 0.83 | 0.79 | 0.79 | 0.79 | 0.72 | 0.72 | 0.72 | 0.72 | 0.71 | 0.71 | 0.85 | 0.85 | 0.85 |
| Total params | 4,700,572.00 | | | 22,245,160.00 | | | 54,668,520.00 | | | 19,105,352.00 | | | 21,697,072.00 | | |
| Trainable params | 4,663,834.00 | | | 22,210,728.00 | | | 54,607,976.00 | | | 18,876,296.00 | | | 21,642,544.00 | | |
| Non-trainable params | 36,738.00 | | | 34,432.00 | | | 60,544.00 | | | 229,056.00 | | | 54,528.00 | | |

computational limitation of the equipment used in this project we safe the metrics and restart the running for each fold, making sure that the data was shuffle for every of them.

The Fig. 6 depicts the confusion matrix performed by different CNN architectures after been tested with 10-fold cross-validation with 44,669 images augmented skin images. The predicted classes are represented as columns while the actual classes are represented as rows. In the diagonal principal the number of hits can be seen, and the intensity of color represents how many hits matched that box.

From the data obtained from the confusion matrix and the formulas explained in "Evaluation Metrics" implemented in the sklearn.metrics library the Table 4 is created. This shows the performance of every trained CNN model.

The Fig. 7 shows the best classification system through the receiver operating characteristics curve (ROC). By obtaining an area under the curve (AUC), the quality of the classifier is evaluated. The closer this area is to the value of one, the better the classifier.

## DISCUSSION

In order to appraise the performance of a modified version of the Nasnet model for discriminating among eight different skin lessons, it is compared against 5 state-of-the-art models, InceptionV3, InceptionResNetV2, DenseNet201, Xception. The models were applied to the ISIC 2019 dataset. Those models are training using 10-fold cross-valid ratio—a novelty in skin lesion classification for the ISIC dataset, based on the reviews recapitulated in the introduction. From the trained models were obtained the confusion

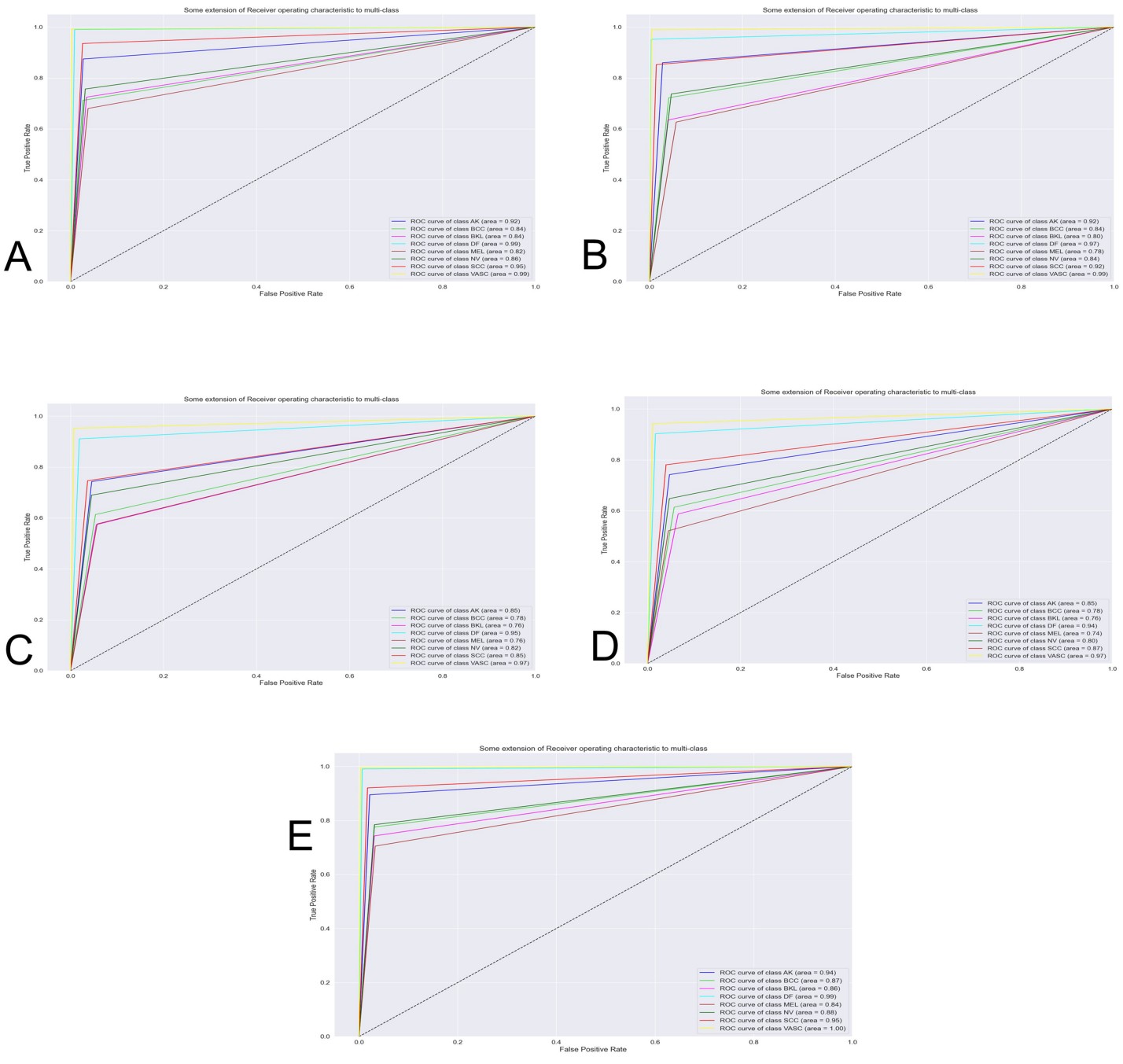

**Figure 7 Curve AUC ROC.** (A) NasnetMobile, (B) Inception V3, (C) InceptionResNetV2, (D) DenseNet201 and (E) Xception. The figure shows the best classification system through the receiver operating characteristics curve (ROC). By obtaining an area under the curve (AUC), the quality of the classifier is evaluated. The closer this area is to the value of one, the better the classifier.

matrix (Fig. 6) and AUCROC (Fig. 7). All these models differ in terms of computational speed (i.e., run time).

On the confusion matrix (Fig. 6), in this case, the proposed model presented a pronounced diagonal principal since most of the predictions were correct. Most of the

classes presented few false positives and false negatives; however, the class Melanoma and Nevus presented more errors due of the visual similarity that these classes share, which is congruent with current, state-of-the-art.

The precision metric points to low performance on the most common type of cancer (melanoma). The confusion matrix makes it evident that the models often confound melanoma and Nevus. This confound is also reported by the dermatologist on their diagnostic of skin disease—leading to the common problem of unnecessary biopsy of a Nevus (*Carrera & Marghoob, 2016*)—which comes from the visual similarity between nevus and melanoma. In future works the results might be improved by adding handcrafted (*Nanni, Ghidoni & Brahnam, 2017*) features to the training process.

Studying the precision in this work brings up an absence of this metric similar reviewed papers, therefore should be encouraged to report this metric to enhance this forgot the aspect of skin lesion classification in further works. The classification report for every trained model is depicted in Table 4. Notably, Nasnet has a similar performance than the best score Xception, with a reduction in the number of trainable parameters of at least 75%. For a future work would be necessary re-training the models on a deep learning workstation and do the comparison after training all the 10-fold at once in order to get a more robust result.

## CONCLUSIONS

This work presents a complete report on the comparison of five different state-of-the-art CNN architectures on the classification task of eight skin lesions (NASNET, InceptionV3, InceptionResNetV2, DenseNet201, Xception). This comparation is establish from the F1-score, Precision, Recall, accuracy and trainable parameters—commonly unreported by similar studies.

The architectures are trained with ISIC 2019 dataset, using 10-fold cross-validation The obtained results could further be increased using segmentation.

All the models get a significant training accuracy based on a dermatologist one who reaches 75% to 84%. The best performance based on the training accuracy is obtained from Xception with 85%. Nevertheless, it was the heaviest to train with at least 5 days of continuing running on the computer. In contrast, the NASNET got the second-best performance with 83% but with a considerably reduction of 75% on the training parameters, also decreasing the running on the computer to just 12 h. The other trained models shows a lower perform than NASNET and Xception.

With the future expansion of ISIC Dataset and another open dataset with dermoscopic pattern annotations become available, future work may consider improve system performance including the use of additional situational contexts, such as patient metadata, history, comparison with other lesions on the patient and evolution through time. In addition, other approaches such as meta-learning, ResNets for semantic segmentation, and complex shape descriptors for classifying diseases might provide additional performance gains. With superior equipment these results may have been more robust and is therefore an area of promising future research.

# ACKNOWLEDGEMENTS

The authors are grateful for the support provided by the National Secretariat of Science and Technology of Panama (SENACYT) and the Technological University of Panama (UTP). Also, we would like to thank the larger community of the International Skin Imaging Collaboration (ISIC) for their effort in organizing the datasets used in this work, as well as engaging and insightful discussions in Dermoscopy and dermatology.

## Funding

This work was supported by the Secretaría Nacional de Ciencia, Tecnología e Innovación of Panama Republic. The funders had no role in study design, data collection and analysis, decision to publish, or preparation of the manuscript.

## Grant Disclosures

The following grant information was disclosed by the authors:
Secretaría Nacional de Ciencia, Tecnología e Innovación of Panama Republic.

## Competing Interests

The authors declare that they have no competing interests.

## Author Contributions

- Elia Cano conceived and designed the experiments, authored or reviewed drafts of the paper, and approved the final draft.
- José Mendoza-Avilés conceived and designed the experiments, authored or reviewed drafts of the paper, and approved the final draft.
- Mariana Areiza performed the experiments, prepared figures and/or tables, and approved the final draft.
- Noemi Guerra analyzed the data, authored or reviewed drafts of the paper, and approved the final draft.
- José Longino Mendoza-Valdés performed the experiments, performed the computation work, authored or reviewed drafts of the paper, and approved the final draft.
- Carlos A. Rovetto conceived and designed the experiments, prepared figures and/or tables, and approved the final draft.

## Data Availability

  The NASNet architecture model is available at Figshare:
  Rovetto, Carlos (2020): NASNet architecture model. figshare. Software.
DOI 10.6084/m9.figshare.13130090.v2.
  The Python code for model training is available at Figshare:
  Rovetto, Carlos (2020): Python code for model training. figshare. Software.
DOI 10.6084/m9.figshare.13130087.v1.

Skin Imaging Collaboration data is available at Figshare:

Rovetto, Carlos (2020): Source Dataset of Skin Imaging Collaboration (ISIC). figshare. Dataset. DOI 10.6084/m9.figshare.13130081.v1.

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
