# Peer review of "Multi skin lesions classification using fine-tuning and data-augmentation applying NASNet"

_PeerJ Computer Science, doi:10.7717/peerj-cs.371_

## Round 0.1 · original submission · Major Revisions

Reviewer 1 explains that the comparison performed is not valid. We have decided not to reject the paper at this point. Still, you need to remake the experimental framework and the comparison of results in the new version of the article to get an acceptance in the next round of reviews.

Reviewer 1 ·

Basic reporting

The goal of the work is to propose an approach using NASNet architecture to recognize eight skin diseases. The paper is in the scope of PeerJ Computer Science. The authors ignore the use of segmentation methods and applied transfer learning and data augmentation. The introduction is extensive literature supported. The authors used a widely adopted data source such as The International Skin Imaging Collaboration . The seven images in the manuscript are low quality.

There is a mistake in the caption and in the header of Table 4.

Experimental design

The authors used 10-fold cross validation, however, Table 4 shows that the authors compared its proposal versus other CNN architectures using different No. classes and evaluation methods. The above comparison is not feasible because all methods have to reproduce the same configuration. The authors should have evaluated the state-of-the-art methods following the same steps, e.g. InceptionV4 needs to be evaluated using the same dataset, the same No. classes and the same 10-fold cross validation. In summary, the comparison performed versus other methods based on Table 4 is not valid and it needs to be performed accordingly the configuration chosen by the authors.

On the other hand, the authors mentioned several metrics to evaluate the performance, but finally they based the conclusions only on accuracy. It should be notice that accuracy is not recommended when there is a high imbalance factor, like the authors recognized in line 208. Also, Figure 6 shows that more than 34,100 images were analyzed during testing, however, the authors already stated that 23,488 are the total number of images, which is inconsistent regarding the evaluation method. In 10-fold cross-validation, for each fold the test set should be composed by approximately 2,348 images. Data augmentation should be applied in such manner that augmented images only compute to find the prediction of the original images, e.g. let us say that test image X is augmented by a factor of 10, then the final prediction of X can be computed using a soft-voting strategy using the 11 images. It should be notice that in this manner the number of test values who computes the final result remains as 2,348.

Validity of the findings

The authors should support the Conclusions based on the evaluation of the methods using the same configuration. In this regard, the experimental study needs to be repeated in some phases in order to sustain the validity of the work.

Reviewer 2 ·

Basic reporting

There are some recommendations and mathematical improvements that I suggest in this paper:
- As a recommendation for a better reading of Section 2, main mathematical expressions and equations should be numbered.

- Equation at the end of line #150: The super-index "l" in the denominator must be "j", to match the index with the summation.

- Beginning of line #151: sigma(z^l) is the probability distribution, not exp(z^l).

- As a recommendation at the end of line #155, "after one-hot" should be substituted by "one-hot encoding".

- Line #156: the numbers of "H1,...,H8" must be subindexes.

- Equation in line #159: Summation indexes must be from k=1 to m.

- English language could be improved in the second part of line #159, "with these notations; the vector...etc". The current phrasing makes comprehension difficult.

- Lines #172 and 173: I suggest justify the selection of mini batches n=30. In fact, the recommendation of your reference [37] says that should be a number between 50 and 256.

- At the end of line #186: In the last equation, "theta" must be a subindex of "nabla".

Experimental design

no comment.

Validity of the findings

no comment.

Additional comments

no comment.

---

## Round 0.2 · Major Revisions

Although this version addresses most of the recommendations given by reviewer 1, reviewer 2 still has major recommendations to give, especially regarding English language quality. I would suggest the authors have the paper reviewed by a native English proofreader.

Reviewer 1 ·

Basic reporting

The authors have fixed the minor corrections, as well the manuscript in overall.

Experimental design

The authors improved the experimental settings and now the manuscript shows extensive results.

Validity of the findings

The results showed Xception achieved 2.4% better validation performance compared to NASNetMobile, however, the authors have a point when comparing by trainable parameters. In this case the best average CNN model was NASNetMobile, as the authors already stated.

There is a concern regarding Table 4. The model achieved the lowest recall performance in Melanoma category. Please notice predicting Melanoma is probably the most important task regarding skin diseases. It will be useful if the authors provide an explanation about the above result.

Reviewer 2 ·

Basic reporting

Correction of English grammar is needed in the whole document.

The conclusions given in Section 3 are not consistent or clearly justified in the rest of the document.

Experimental design

There is a lack of clarity and high technical descriptions.

As a recommendation for a better reading of Section 2, the mathematical expressions and equations should be numbered.

Validity of the findings

The conclusions given in Section 3 are not consistent or clearly justified in the rest of the document.

Additional comments

- Correction of English grammar is needed in the whole document.

- Line #44: ID of melanoma is missing.

- As a recommendation for a better reading of Section 2, the mathematical expressions and equations should be numbered.

- Line #150: Equation at the end of this line: The super-index "l" in the denominator must be "j", to match the index with the summation.

- Beginning of line #151: sigma(z^l) is the probability distribution, not exp(z^l).

- Line #154: "x8" must be "m" or another literal because it represents the number of training data, not the number of classes.

- Line #155, "after one-hot" should be substituted by "one-hot encoding".

- Line #156: the numbers of "H1,...,H8" must be subindexes.

- Equation in line #159: Subindex of summation must be "k", not "m".

- Correction of English grammar in lines #159 to 161.

- Beginning of the line #161: subindexes "j" and "k" are missed in the weight "wl".

- Equation in line #162: the sigma/summation letter is not needed in the compact vectorized form.

- Lines #172 and 173: I suggest justifying the selection of mini batches n=30. In fact, the recommendation of your reference [37] says that it should be a number between 50 and 256. So, why did you use 30?

- Line #174: "975 steps: You need to mention the size of your dataset: training, validation and test sets.

- Line #174: "respited"? Did you mean "repeated"?

- Line #185: "Where gi to denote the gradient i at time step t" must be:"where gt denotes the gradient at time step t".

- At the end of line #186: In the last equation, "theta" must be a subindex of "nabla".

- Lines #195 to 202: A detailed description of the dataset must be included: the size of the dataset and a detailed description of each variable, the number of classes of the dataset: training, validation and testing.

- Line #207: an extra line was included.

- Line #218: Correction of English grammar is needed.

- Line #219: What is the "small number of images" that you are talking about?

- Line #273: Table 2 is wrong. For example, at least the values of the main diagonal must be TP. Include who are the actual values and the predictions. And denote the explicit number of classes, 8, not just as "j".

- Line 288: "aF1 Score" must be "F1 Score".

- Line #312: What are the sizes of training, validation, and testing?

- Line #313: What is the original dimension of each image?

- Line #315: A detailed description of the classes imbalanced must be included.

- Line #326: The size of the partitions must be clarified: in Line #312 you are talking of 23,488 images; in Line #326 you are talking of 44,669 augmented images; and the confusion matrix of Figure 6 has around 41,000.

- Line #330: Not only Melanoma and Nevus had significant errors, from the confusion matrix of Figure 6, we can observe that AK, BCC, and BKL have equal or higher errors than ML and NV.

- Line #334: The number of classes/diseases is 8, but the text in Figure 6 is talking of 9 classes. Why is this?

- Line #334: The notation for the skin classes used in Line #155 and the text of Figure #6 must be the same.

- Line #334, Figure 6: Confusion Matrix: To have a better idea of the magnitude of the errors is recommended to include another confusion matrix, but with percentage values.

- Line #338. 83% for all the metrics used? That is very odd. And at least from the values that we can see in the confusion matrix of Figure 6, the accuracy is approximately 84.3%.

- Line #349. It must be Table 4.

- Line #351: The direct comparison of your method with [30] in Table 4 does not have too much sense, because [30] is using more classes, 10, and your method is using 8 classes.

- In Line #330 you are saying that Melanoma and Nevus have the higher errors (as we can observe from the confusion matrix of Figure 6) because the visual similarity, but now your conclusion in Line #359 is that you can discriminate them? How is that?

- Lines #359. And what about Vascular (VASC) and Dermafibroma (DF)? Confusion matrix of Figure 6, shows them with the best results.

- Line #361: These values, of accuracies and losses, are not clearly justified in the document.

- The conclusions given in Section 3 are not consistent or clearly justified in the rest of the document.

---

## Round 0.3 · accepted · Accept

Good work! This last version is a significant improvement: the reviewer finally decided to support the acceptance of your paper.

Congratulations!

Reviewer 1 ·

Basic reporting

The authors have improved the manuscript in overall.

Experimental design

The experimental study was already extended in the previous revision.

Validity of the findings

The authors provided an explanation about the performance by comparing to dermatologists and achieving similar accuracy.